# Evaluation of NADES for Pectin Films Reinforced with Oxalic Acid-Modified Chitin Nanowhiskers

**DOI:** 10.3390/polym17050572

**Published:** 2025-02-21

**Authors:** Andrea Mathilde Mebert, Cynthia Melisa Melian-Queirolo, Maria Fernanda Hamet, Guillermo Javier Copello, Andrea Gomez-Zavaglia

**Affiliations:** 1Universidad de Buenos Aires, Facultad de Farmacia y Bioquímica, Departamento de Ciencias Químicas, Junín 954, Buenos Aires C1113 AAD, Argentina; amebert@ffyb.uba.ar (A.M.M.); cyn.m.melian@gmail.com (C.M.M.-Q.); gcopello@ffyb.uba.ar (G.J.C.); 2Instituto de Química y Metabolismo del Fármaco (IQUIMEFA), Consejo Nacional de Investigaciones Científicas y Técnicas (CONICET)-Universidad de Buenos Aires, Junín 954, Buenos Aires C1113 AAD, Argentina; 3Center for Research and Development in Food Science and Technology (CIDCA, CCT-CONICET), La Plata 1900, Argentina; fernandahamet@gmail.com

**Keywords:** pectin, natural deep eutectic solvent, chitin nanowhisker

## Abstract

The effect of three NADESs as pectin film plasticizers was evaluated at 10%, 30%, and 50% *w*/*w* by using the casting method. Two hydrophilic (choline chloride with glycerol or citric acid) and one hydrophobic (thymol–camphor) NADESs were used as replacement for glycerol. Oxalic acid-modified chitin nanowhiskers (oCNWs) at 1% *w*/*w* were used to evaluate the effect of the NADESs on the nanofiller. The resulting films using the hydrophobic NADES were difficult to handle and prone to cracking and performed similarly to or worse than pure pectin films. As a result, they were not further evaluated. In contrast, the hydrophilic ones showed characteristics comparable to glycerol. It was found that films containing glycerol and choline chloride–glycerol NADESs showed a decrease in opacity and tensile strength and an increase in WVP, Young’s modulus, and maximum elongation. In contrast, those that contained citric acid exhibited a different behavior: opacity was less affected, and a decrease in WVP and an increase in tensile strength and Young’s modulus (at 10% and 30% plasticizer) were found. oCNWs tended to decrease WVP and increase Young’s modulus but not in a very significant way. Our findings demonstrate that NADESs can be used as plasticizers in pectin films without the need to include glycerol and that the nature of NADESs is relevant to tuning the final properties.

## 1. Introduction

Pectin is a biopolymer widely distributed among vegetables that can be abundantly obtained, including from agro-industrial waste. Pectins are structurally composed of linear chains of partially methyl-esterified (1→4)-linked α-d-galacturonic acid units, which may also include neutral sugars like glucose or galactose. Based on their degree of methoxylation (DM), pectins are categorized into high-methoxyl pectins (>50% esterification) and low-methoxyl pectins. The DM significantly influences their functional and technological characteristics. This supports their promising use as an alternative to petrochemical plastics in the food packaging industry. In spite of that, pure pectins exhibit physicochemical characteristics that are unsuitable for this purpose. Therefore, efforts are focused on enhancing film-forming properties with additives such as cross-linkers, plasticizers, and nanofillers, among others [1].

Glycerol has been widely used as a plasticizer for many biopolymer films with optimal performance. Nevertheless, studying plasticizers with varied chemical moieties can tune the properties of the resulting material through specific interactions between the polymer matrix and the plasticizer’s moieties. Therefore, focusing on evaluating different natural deep eutectic solvents (NADESs) for their use as potential green plasticizers is of great interest. NADESs are formed by mixing hydrogen bond donors (HBDs) and hydrogen bond acceptors (HBAs). Within NADESs, there is a specific subcategory of natural deep eutectic solvents where both the HBDs and HBAs consist of primary metabolites, considered human- and environmentally friendly, i.e., amino acids, organic acids, sugars, or choline derivatives [2].

One advantage of using NADESs as plasticizers is their ability to not only intercalate between polymer chains but also form strong interactions through their chemical groups. These solvents have been extensively studied as plasticizers in other biopolymers, such as chitosan [3] and starch [4]. Although there are many studies exploring the potential use of NADESs to extract pectin or aid pectin extraction [5,6,7], there are few embracing their use as plasticizers. Moreover, many of them use glycerol-containing NADESs, but there are a wide variety of NADESs to be explored [8,9,10]. Thus, this could present a potential future advantage by enabling the direct use of the extraction solvent as a plasticizer, thereby reducing the number of material processing steps.

Nanofillers are another approach to improving the functionality of nanocomposite materials with biopolymers, for the improvement in film performance in terms of mechanical behavior and barrier properties and even for specific functionalities, such as antimicrobial properties [11]. Many nanofillers have been studied in biopolymers films for packaging applications [12,13]. The innocuity of the filler becomes a critical limitation. Increasing concerns about the migration of metal nanoparticles in food-contact materials and their associated health risks have sparked discussions on the safe use of nanomaterials. In this sense, nanofillers derived from natural food-grade biopolymers (i.e., starch, cellulose, chitosan, and proteins) present a promising solution paving the way for the creation of nanocomposites designed for safe and sustainable food packaging [14,15,16,17]. Chitin- and chitosan-based nanofillers are considered biocompatible and biodegradable and to have low toxicity [11].

In this work, pectins from citrus were used to prepare films incorporating oxalic acid-modified chitin nanowhiskers as nanofillers and NADESs as plasticizers. The resulting films were characterized by using microscopic, mechanical, and analytical techniques (FT-IR, SEM, Vis spectrometry, Water Vapor Permeability, and uniaxial traction), providing essential insights for sustainable applications in food technology. Choline chloride-based NADESs with glycerol or citric acid were used. Our findings demonstrated that NADESs act as plasticizers without the need to contain glycerol. Moreover, the behavior of the NADES containing citric acid was different in many aspects, which indicates that the composition of the NADES influences the final properties of the obtained film, making it a point of interest for tuning its properties.

## 2. Materials and Methods

### 2.1. Reagents

Pectin from citrus peel (SIGMA-ALDRICH, Søborg, Denmark) with degree of methoxylation of 48.5% (analyzed by FT-IR [18]), chitin from shrimp shells (SIGMA, Livonia, MI, USA), glycerol (Stanton, Buenos Aires, Argentina), choline chloride (Acros Organics, Shanghai, China), anhydrous citric acid (Stanton, Argentina), thymol, camphor, and oxalic acid dihydrate (Stanton, Argentina) were purchased from a local supplier.

### 2.2. Synthesis of Chitin Oxalic Acid-Modified Nanowhiskers

Chitin oxalic acid-modified nanowhiskers (oCNWs) were prepared as follows: A total of 20 g of oxalic acid and choline chloride NADES (2:1 molar ratio) was prepared at 100 °C under constant stirring. After a homogeneous liquid was formed, 400 mg of chitin was added. It was left at 100 °C under stirring for 1 h. Immediately after this time lapse, the mixture was removed from the heat bath, and 20 mL of deionized water was added. It was consecutively washed three times with 20 mL aliquots of water through centrifugation, then sonicated at room temperature at 35 kHz for 30 min, and washed again three times with aliquots of 20 mL of water. Finally, it was resuspended in 20 mL of the same solvent and stored in the refrigerator until use (4 °C) (Figure 1A). Non-modified whiskers were prepared by basic hydrolysis by using 0.1 M NaOH over 30 min.

### 2.3. Synthesis of Natural Deep Eutectic Solvents

Natural deep eutectic solvents (NADESs) were prepared as in the literature. Briefly, adequate proportions of each HBD and HBA were mixed in a heating bath under moderated stirring until a clear solution was observed and then were cooled down to room temperature (Figure 1B). The following mixtures were prepared at the proper temperatures: glycerol and choline chloride (GC; 3:1 molar ratio) at 80 °C [19] and citric acid and choline chloride (CC; 1:2 molar ratio) at 50 °C, after a homogeneous solution was formed. Then, 30% *w*/*w* of deionized water was added [20]. Thymol and camphor (TC; 1:1 molar ratio) were prepared at 50 °C [21].

### 2.4. Pectin Film Formation

The pectin films were prepared by the casting method. Briefly, 1 g of pectin powder was mixed with 300 mg of plasticizer (either glycerol or NADES) or water in the control case. It was manually homogenized; then, 30 mL of deionized water was added in aliquots of 5 mL while stirring. The mixture was covered and left overnight at room temperature under moderated stirring. Then, it was centrifuged at 6000 rpm for 1 h to avoid bubbles, poured into a suitable mold (13 × 9.5 cm), and left to dry at 37 °C for 48 h. After this time, the formed films were demolded (Figure 1C). For films containing nanowhiskers, 10 mg of oCNWs was dispersed in 300 mg of the plasticizer. The final proportions were 30% plasticizer and 1% nanowhiskers with respect to the weight of the polymer.

Films with 10% and 50% plasticizer were prepared in the same way but by adding 100 or 500 mg instead of 300 mg of plasticizer.

### 2.5. Infrared Spectroscopy

ATR-FTIR (diamond attenuated total reflectance) spectra were recorded by using a Nicolet iS50 Advanced Spectrometer (Thermo Scientific, Miami, FL, USA). The ATR-FTIR spectra were recorded with 32 scans at a resolution of 4 cm^−1^.

### 2.6. Film Thickness, Length, and Width

Film thickness was measured at at least 10 random different points with a digital coating thickness meter (Reed CM-8822 Coating Thickness Gauge) for traction tests and with an Electronic Syntek Digital Micrometer in any other test. Length and width were measured with a Digital Caliper Hamilton C30.

### 2.7. Transmission Electron Microscopy

The size and the shape of the oCNWs were investigated by Transmission Electron Microscopy (TEM). Briefly, a drop of the sample in aqueous solution was deposited on 400-mesh carbon-coated copper grids. The dried sample was negatively stained with phosphotungstic acid. TEM was performed at room temperature by using a Zeiss 109 electron microscope (Zeiss, Oberkochen, Germany).

### 2.8. Scanning Electron Microscopy

The morphological characterization of the obtained film surfaces was performed by using a Philips 505 microscope (Philips Industries Eindhoven, The Netherlands). Each film was dried and cut into two pieces with a scalpel. Then, the samples were coated with 20 nm of gold before obtaining Scanning Electron Microscopy (SEM) images (15.0 keV).

### 2.9. Film Transparency and Opacity

The optical properties were evaluated by spectrophotometry as recommended in the work by Liu et al. [22]. Transmittance and absorbance were determined at 600 nm by using a SP-2000UV UV-Vis Spectrophotometer (Spectrum, Shangai, China). The transparency and the opacity were determined as
Transparency = %T_600nm_ = 10^(2-A)^(1)
(2)$$Opacity=\frac{A600\mathrm{nm}}{X}$$
where *X* represents film thickness in mm.

The assay was performed in triplicate.

### 2.10. Water Vapor Permeability Assay

Water Vapor Permeability (WVP) tests were performed according to ASTM Standard Test Method E96 [23]. Briefly, a vial was filled by 3/4 with water and sealed with the film to be evaluated. Then, it was placed in a dry, sealed compartment full of dried silica gel and further incubated at 35 °C. The exposed surface area of the material interconnecting both compartments was 0.00054 m^2^. The relative humidity in the inner compartment was considered 100%, while that on the opposite side, 0%. The vial was weighed at different time intervals (two times per day for 4 days), and the water loss was calculated in g/s. A water vapor pressure differential value of 5626.45 Pa (35 °C) [24] was used for assuming full water vapor saturation in the headspace and a fully dried environment provided by the silica gel. WVP was calculated as follows:(3)$$WVP=\frac{dW*X}{dt*A*dP}$$
where d*W* and d*t* represent the change weight (g) in the time lapse (s), *X* is the film thickness (m), *A* is the film’s area exposed (m^2^), and d*P* is the vapor pressure differential (Pa).

A blank of each type of material was performed by using an empty vial. The assay was performed in triplicate for each film and blank.

### 2.11. Uniaxial Traction Test

Tensile tests were carried out in a TA.XT2i Texture Analyzer (Stable Micro Systems, Surrey, UK), using a crosshead speed of 1.0 mm/min and a load cell of 0.1 kN. Tensile strength (σm), Young’s modulus (E), and elongation at break (εB) were obtained by following ASTM D1708-18 standard [25] recommendations. Probes of 9.5 cm in length and 0.75 cm in width were cut. At least 10 films were evaluated.

### 2.12. Statistical Analysis

In all cases, the data are presented as means ± SDs. One-way ANOVA with Tukey’s post-test was performed by using GraphPad Prism version 5.00 for Windows, GraphPad Software (San Diego, CA, USA); *p* < 0.05 was considered significant.

## 3. Results

### 3.1. Nanowhisker Characterization

Due to the neutral nature of chitin, CNWs tend to agglomerate, leading to heterogeneous materials when they are used for film reinforcement. The addition of ionizable moieties, such as -COO^−^, aids the dispersibility of the CNWs in the casting solution, in turn facilitating the obtaining of homogeneous materials. The surface modification of CNWs can be performed after their synthesis, but the NADES synthesis method has the advantages of both being green and achieving modified CNWs in one step, thus being a preferred method of choice. The synthesized oCNWs were characterized by TEM and FT-IR spectroscopy (Figure 2a,b). In the TEM images, nanowhiskers of around 10 ± 5 nm in diameter can be seen. In FT-IR, the spectra denoted the formation of a new peak at 1734 cm^−1^, which is related to the formation of the ester group. This peak is absent in pure chitin powder and non-esterified CNWs. This peak disappeared after basic hydrolysis.

### 3.2. Natural Deep Eutectic Solvent Characterization

Proper NADES formation was checked by FT-IR, as this technique can be effectively utilized to characterize the H-bond interactions occurring in these DES formations. The GC-NADES spectrum shows the characteristic peaks of the disappearing of the OH stretching peak of choline chloride at 3220 cm^−1^ linked to the choline chloride OH(Ch^+^)–Cl^−^ hydrogen bond, as well of the apparition of characteristic absorptions from the (CH_3_)_3_N^+^ group of choline chloride at 3027 and 1478 cm^−1^, among others [26] (Appendix A). In the CC-NADES, usually, a shift in O-H choline chloride stretching is related to proper DES formation, but due the presence of water in this band causes a shift in other signals; for instance, the C-O stretching of citric acid (1204 cm^−1^ to 1197 cm^−1^) and the C-N stretching (1084 cm^−1^ to 1081 cm^−1^) of choline chloride were also used [27] (Appendix A). The TC-NADES spectrum shows a broadening of the O-H stretching vibration band of thymol at 3194 cm^−1^. This alteration indicates a decrease in the force constant, likely caused by proton transfer through hydrogen bonding, which results in the broadening of the peak [28] (Appendix A).

### 3.3. Pectin Films

Films circa 0.07 mm thick were prepared by the casting method. The thickness tended to increase with the addition of more plasticizer mass (Appendix A). Loads of 10%, 30%, and 50% of different plasticizers were tested. Glycerol was used as the control, as it is commonly used as a biopolymer plasticizer [29]. Two hydrophilic NADESs were tested with choline chloride as the HBA and glycerol or citric acid as the HBD. A hydrophobic NADES containing thymol as the HBA and camphor as the HBD was also tested. The films obtained with the hydrophobic NADES broke easily when handled, showing themselves to be as brittle as the pure pectin ones. Thus, they were not studied further.

#### 3.3.1. FT-IR and SEM Characterization

The pectin esterification degree was analyzed by FT-IR. A DM value of 48.5% was found by relating the C=O and COO- signals at 1739 and 1622 cm^−1^, respectively [18]. This means that there were almost 50–50 esterified and free carboxylic groups, although it is low-DM pectin (<50%).

The changes in the infrared spectrum of pure pectin films were analyzed to confirm the incorporation of the plasticizers and verify possible interactions between the plasticizer and the polymer. The presence of oCNWs and plasticizers at 10% in all cases showed no significant changes when compared with the spectrum of pure pectin films. Compared with the spectra of pure pectins, those corresponding to films containing 30% and 50% glycerol or GC-NADES showed an increase in the intensity of the peaks in the 1100–900 cm^−1^ region. Pure glycerol-plasticized films presented many peaks in this region, and their relative intensity increased with the plasticizer. There were no evident signs of interaction, as these changes may be due to spectral addition, but it confirms the incorporation of different quantities of the plasticizer.

When using the GC-NADES or CC-NADES, the 958 cm^−1^ peak increased (along with a slight shift) as the plasticizer percentage rose, which may be due to the spectral overlap with the 953 cm^−1^ peak of choline chloride. Another indication of the presence of NADESs was the peak at 1481 cm^−1^, which was absent in pectin but present in choline chloride, and it increased as the proportion of the GC-NADES or CC-NADES increased. As the concentration of the CC-NADES plasticizer increased, the relative intensity of the pectin peak at 1077 cm^−1^ increased, matching that of the 1095 cm^−1^ peak at the highest plasticizer concentration. The relative intensity of the pectin peak at 1622 cm^−1^ decreased compared with that at 1739 cm^−1^, but this change in relative intensity was similar for both concentrations, showing no evident relationship with the amount of plasticizer. Therefore, it did not seem to be related to a simple spectral summation. As previously mentioned, these peaks are associated with COO^−^; and C=O, which could suggest an interaction between citric acid and the carboxylic groups of the polymer (Figure 3).

By SEM imaging, some differences in the surfaces of the films were observed (Figure 3). For example, in those made of pure pectin (Figure 4(aii)), the surface appeared smoother than that of those containing oCNWs (Figure 4(aiv)). The surfaces of those materials plasticized with 10% plasticizers appeared smooth. Those containing glycerol or the GC-NADES in amounts of 30% or 50% showed cracks in many sections of the material. In the case of the CC-NADES, these cracks were only observed when using 50% and to a lesser extent compared with the other two plasticizers.

#### 3.3.2. Optical Properties

The films’ transparency and opacity were evaluated as recommended in an interesting work by Zhao et al. [22]. Film transparency, in most cases, was not altered by adding plasticizers nor oCNWs. On the other hand, while evaluating the opacity of the films, which relates to the absorbance at 600 nm and film thickness (mm), a trend was observed. The addition of plasticizers decreased, in most cases, the films’ opacity, to circa one-third of that of the pure pectin films. The presence of oCNWs slightly increased opacity but not in a significant way (Figure 5a,b and Appendix A).

#### 3.3.3. Water Vapor Permeability

The WVP of the films was evaluated with 0% RH outside and 100% RH in the inner compartment. While films with pectin with oCNWs, 10% plasticizers, and 30% CC-NADES showed no significant difference from pure pectin films, the addition of 30% glycerol and GC-NADES increased WVP by around two times. Films with 50% of these plasticizers showed a higher value of WVP than 30% films, but this was not significantly different. When adding 50% CC-NADES, the WVP value increased but to a lesser extent than when adding 30% and 50% glycerol or CG-NADES. When oCNWs were dispersed in 30% plasticizers, it seems to lower the WVP compared with 30% films without nanowhiskers but not in a significant way (Figure 6a and Appendix A).

#### 3.3.4. Tensile Test

The Young’s modulus, maximum tensile strength, and elongation (at break point) of the films were evaluated by the uniaxial traction test. Most of the pectin, pectin oCNWs, 10% GC-NADES probes broke before testing. Only the values for which at least triplicates could be obtained were considered; for this reason, the data obtained from pectin oCNWs and 10% GC-NADES are not presented.

Mainly, the addition of plasticizers, pure glycerol, and the GC-NADES lowered the maximum tensile strength. This decrease was more marked in glycerol-plasticized films and when the amount of plasticizers was 50%. A similar tendency was observed in Young’s modulus. The CC-NADES films at low concentrations (10% and 30%) showed the opposite tendency: greater values of tensile strength and Young’s modulus were observed. All materials showed greater tensile elongation than pure pectin films; this value increased with the increase in plasticizers, and pure glycerol showed greater values than both NADESs. The presence of oCNWs did not seem to impact the maximum tensile strength and the maximum elongation, but it caused a little increment in Young’s modulus with respect to the films containing the same amount of plasticizer (Figure 6b–d and Appendix A).

## 4. Discussion

Glycerol was chosen as the model plasticizer because it is the most widely used for biopolymers and would allow for a comparison with other materials previously reported [29]. As NADESs, two hydrophilic options were selected, i.e., one based on glycerol and choline chloride (3:1), and another with citric acid and choline chloride (1:2), to assess the presence of glycerol in the solvent as a crucial factor while having the same HBA. Plasticizers were evaluated in loads of 10%, 30%, and 50%. Although the literature varies widely in the amounts used, a quantity of 30% is commonly accepted as a midpoint load. In this work, modified chitin nanowhiskers (oCNWs) were also evaluated as nanofillers. oCNWs were added at 1% to films containing 30% plasticizer (midpoint), as this concentration was found to modify the properties of many polysaccharide films [30]. Whiskers, also known as crystalline nanofibrils, are materials derived from the breakdown of crystalline structures into nanocrystalline forms with defined shapes or through the self-assembly of fundamental building blocks. Chitin nanowhiskers have a reactive surface rich in hydroxyl groups, allowing for potential modifications [31], which were esterified with oxalic acid.

The casting method was successfully used for preparing pectin films. As expected, the implementation of plasticizers was necessary to obtain ductile films. Remarkably, only the hydrophilic plasticizer led to flexible films, whereas the hydrophobic NADES led to brittle ones. Although a hydrogen bond donor/acceptor behavior among the component of the hydrophobic NADES is described in the literature, probably, the remaining moieties in its component are not able to interact with the hydrophilic pectin moieties by hydrogen bond or electrostatic interactions.

Infrared spectroscopy was useful to confirm the production of esterified CNWs, as well as for evaluating the correct formation of the NADESs. This technique demonstrated the presence of different amounts of plasticizer in the films through the increased intensity of specific signals. Moreover, the signal shifts and intensity variations observed for several peaks, compared with those in the spectrum of the pure pectin film, provided evidence of the type of interaction between the plasticizer and the biopolymer. For example, the interaction between the acidic groups of citric acid and the non-esterified groups of pectin was evidenced. The relative intensity of the pectin peak at 1622 cm^−1^ (COO^−^) decreased compared with that at 1739 cm^−1^ (C=O). However, unlike the signals that showed a more significant increase in intensity with higher CC-NADES plasticizer content, this signal was similarly modified in the films containing 30% and 50% plasticizer. This makes it more challenging to ascribe the change to spectral addition and could instead be linked to the interaction between the acidic groups of citric acid and the non-esterified groups of pectin.

By SEM imaging, some differences in the surfaces of the films were observed. For example, in those made of pure pectin, the surface appeared smoother than that of those containing oCNWs. This could indicate that using water as a solvent to integrate the oCNWs into the pectin did not allow for proper homogenization. Also, the oCNWs may have a great influence on pectin structuring due to direct interaction with the polysaccharide when no plasticizer is used. Another finding was that adding plasticizers resulted in small cracks observed in several sections of the films when their load was over 10%. This was more evident when glycerol was used as a plasticizer and occurred to a lesser extent when the CC-NADES was used, being noticeable only at the highest concentration (50%) and to a lesser degree than with the other two plasticizers. This could imply that the plasticizers migrate to the surface of the film and affect the gold coating applied for SEM imaging.

The optical properties of the pectin films were evaluated by Vis spectroscopy. A value of transparency of 83%T and a value of opacity of 1.46 Abs/mm at 600 nm for pure pectin films was found. In this work, the suggestions by Zhao et al. [22] were followed, expressing transparency as %T and opacity as Abs/mm at 600nm. In addition, other values expressed as %T/mm are provided in Appendix A, as this format has been used in other works and could facilitate comparisons. In most cases, the transparency of the film (%T) showed no significant differences, and no clear trend was observed with the addition of either the plasticizer or oCNWs. This observation is consistent with findings from other studies, such as cellulose–glycerol mixtures (with up to 50% glycerol), where no significant differences in transparency were observed among films with varying amounts of plasticizer, all of which were classified as highly transparent [32].

A decrease in opacity (Abs/mm) was observed with the addition of plasticizer, and this reduction became more pronounced as the amount of plasticizer increased. Opacity was lower when glycerol was used as the plasticizer, but when the CC-NADES was used as the plasticizer, this parameter was less affected. It was reported that in protein films, the addition of glycerol reduces opacity compared with pure film [33]. Pure pectin films presented a value of 1.461 Abs/mm, while this value dropped significantly by adding glycerol (1.081 Abs/mm, 0.907 Abs/mm, and 0.705 Abs/mm, for 10%, 30%, and 50%, respectively) and less markedly with the GC-NADES (1.139 Abs/mm, 0.956 Abs/mm, and 1.062 Abs/mm, for 10%, 30%, and 50%, respectively). The opacity of 30% G films of 0.9 Abs/mm found was comparable to that of 30% G-CaCl_2_ of 0.8 Abs/mm reported in another work [34]. The addition of oCNWs to the films seems to increase the opacity value in most cases: pectin with 1% oCNWs (1.564 Abs/mm), 30% glycerol with 1% oCNWs (0.961 Abs/mm), and 30% GC-NADES with 1% oCNWs (1.262 Abs/mm). The influence of the CC-NADES on the opacity values was not so clear, as addition at 10% did not affect this parameter in a significant way (1.517 Abs/mm), 30% addition lowered it (1.165 Abs/mm) (but less than the other plasticizers), and 50% showed a small increase (1.804 Abs/mm) in opacity compared with pure pectin films. Although variations in the opacity values among films of the same thickness were found, thickness could be a factor influencing this parameter and should be considered for the correct interpretation of the results.

In the present work, a value of 0.203 g/s·m·Pa WVP for pure pectin was found, and the WVP value increased as glycerol and the GC-NADES plasticizer were added in proportions greater than 10%. Values of 0.388 g/s·m·Pa and 0.398 g/s·m·Pa were found for 30% and 50% glycerol. This increase was somewhat less pronounced when glycerol mixed with choline chloride in the GC-NADES was used, with the increases being 0.330 g/s·m·Pa and 0.377 g/s·m·Pa for 30% and 50% NADES, respectively. Probably, the high content of glycerol in the GC-NADES led to a similar behavior in these films. A similar trend was reported, with WVP increasing as the amount of glycerol increased in glutaraldehyde cross-linked pectin films [35]. While similar tendencies in WVP were reported, values with high variability can be found in the literature. Differences can be ascribed to pectin sources and its DM rate, different film production methods, additives, and sample storage prior measurements. For example, lower WVP values of 0.021 to 0.041 g/m.s.Pa (1.27–2.47 g.mm/Kpa.h.m^2^) for pure pectin with high-methoxyl pectin obtained by extrusion/thermo-compression were reported [36]. In our work, 30% G films presented a WVP value of 0.388, while for 30% G films with similar thickness and production method but treated with calcium chloride solution, a value of 13 × 10^−11^ g/m.s.Pa was reported (pectin at ≥74%, methoxyl groups) [34]. The tested conditions also affect the obtained values of WVP, as it has been reported that in alginate films, the WVP values were greater under the tested condition (100% RH–0% RH) than at 76% RH–0% RH. And films with plasticizers (using fructose, glycerol, sorbitol, and polyethylene glycol (PEG-8000)) tend to absorb more water [37]. So, the obtained values are not easily comparable to the ones reported previously by other authors. The incorporation of glycerol induces structural modifications in the polymer network, reducing its density and enhancing its flexibility. These alterations generate additional pathways for water vapor diffusion, leading to an increase in WVP [38,39]. Increased WVP when using NADES plasticizers was expected, as the ability of gelatin films with the GC-NADES (2:1) at 10% and 30% to regulate moisture variations has been reported. GC-based films exhibited moisture absorption capacity and an improved effective diffusion coefficient. Due to structural similarities between DES-based films and buffer pools, the mechanism of water flow regulation in buffer pools resembles the water molecule transfer process in these films [40]. Interestingly, a different behavior was observed in the glycerol-free CC-NADES; a decrease in the WVP value was observed at 30% (0.163 g/s·m·Pa), and at 50%, an increase (0.278 g/s·m·Pa) was noted, but it was significantly less pronounced compared with other plasticizers and significantly different from them (*p* < 0.005 for 50% glycerol and *p* < 0.001 for 50% GC-NADES). This can be related to the cracks observed in the SEM images; in those materials where WVP increased, cracks were more evident.

In this study, 30% films without oCNWs showed higher values of WVP than those with nanofillers. Although it is widely described that nanofillers increase the tortuosity of the gas path and therefore tend to reduce WVP, the fact that, in this case, this reduction was not statistically significant could be due to the increase in more varied interactions of the NADESs with the pectin structure, as also seen in the films’ mechanical behavior as mentioned below. The trend of oCNW nanofiller was similar to that of reported unmodified chitosan nanoparticles in pectin films, which was shown to lower WVP in low-methoxyl pectin [41].

A maximum tensile strength of 34 MPa, a Young’s modulus of 7.9 MPa, and 2.7% maximum elongation in pure pectin films were found. The addition of plasticizers in all cases lowered the maximum tensile strength. This decrease was more pronounced in those containing glycerol and less marked in those including the CC-NADES, showing values of 17.7 MPa, 23.4 MPa, and 31.8 MPa for glycerol, GC-NADES, and CC-NADES, respectively. It has been reported that the addition of glycerol results in weaker, more deformable, and flexible films compared with pure pectin. This effect is attributed to an increase in the amorphous phase of the films, which becomes more pronounced as the glycerol content increases [35]. The maximum tensile strength observed with the addition of 10% and 30% CC-NADES was higher than that of pure pectin. As discussed before, values are not easy comparable, as many factors alter tensile results; for example, it was reported that as the relative humidity increased, a decrease in tensile strength and increased elongation were observed [37]. Tensile strength between 3.44 and 8.74 MPa in high-methoxyl pure pectin films obtained by extrusion/thermo-compression [36] and 3270 MPa glutaraldehyde cross-linked pectin films were reported [35].

Young’s modulus exhibited a similar trend to the maximum tensile strength, decreasing significantly when 50% plasticizer was added, with values of 2.7 MPa, 5.0 MPa, and 6.4 MPa for 50% glycerol, 50% GC-NADES, and 50% CC-NADES, respectively. However, a slight increase was observed with 10% (17.7 MPa) and 30% (11.4 MPa) CC-NADES. The maximum elongation at break also increased with the addition of plasticizers, with higher elongation values observed as the plasticizer content increased. The observed increase in elongation was more significant with glycerol, an increase of about 500% in relation to pure pectin films, whereas the GC- and CC-NADESs showed the smaller increases (around 200% and 300% in relation to pure pectin films). The percentage changes relative to the blank are shown in Appendix A.

The addition of oCNWs did not modify the maximum tensile strength of the films. Young’s modulus increased in the presence of oCNWs, which was expected. The addition of nanowhiskers did not produce a clear effect on the maximum elongation; in the case of glycerol, a decrease in this parameter was observed, while with the other plasticizers, no change or a slight increase was noted. The elongation at break is typically reduced by the addition of nanofillers, although there are reports, for example, on natamycin-loaded zein/casein composite nanoparticles, where this parameter was increased [42]. Again, it seems that NADESs’ interaction with pectin hinders the expected effect of the nanofiller.

## 5. Conclusions

The films produced with hydrophilic plasticizers were easily manipulable, bendable, and easy to cut, unlike the pure pectin films and the ones containing hydrophobic NADES. It was found that these NADESs acted as plasticizers in a manner comparable to glycerol, without necessarily containing it. The glycerol-based plasticizer tends to affect the evaluated parameters in a similar manner to pure glycerol but generally in a less pronounced way. The characteristics of the HBA and the HBD have a relevant effect on their plasticizing behavior. The combined effect of NADESs and oCNWs showed to be less pronounced when compared with glycerol and nanofillers.

The combination of plasticizers and nanofillers proves to be a promising strategy, successfully modifying the properties of pure pectin and making them tools for its tunability. Our results demonstrate that films with additives that pose low risk for human health and are environmentally friendly, such as NADESs and oCNWs, are promising for use in food packaging, potentially serving as a viable alternative and addressing sustainability and human health safety issues.

## Figures and Tables

**Figure 1 polymers-17-00572-f001:**
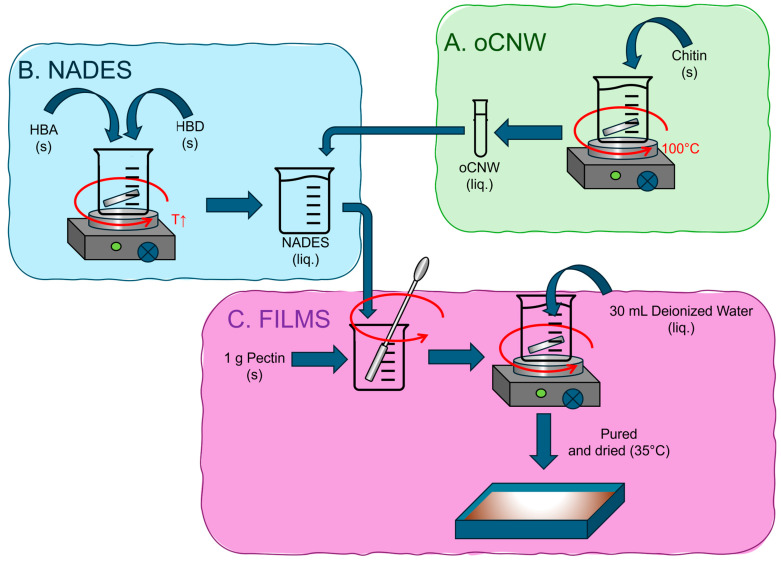
Synthesis scheme. (**A**) oCNW synthesis: oCNWs were prepared by heating chitin in oxalic acid and choline chloride NADES (2:1) at 100 °C 1 h and washed. (**B**) NADES preparation: HBAs and HBDs were heated under constant stirring until a homogeneous liquid was obtained. When the nanofiller was used, a volume containing 10 mg of oCNW was added to 300 mg of plasticizer. (**C**) Film synthesis: A total of 300 mg of plasticizer (30%) or water (pure pectin films) was placed in 1 g of pectin, mechanically homogenized, and allowed to dry.

**Figure 2 polymers-17-00572-f002:**
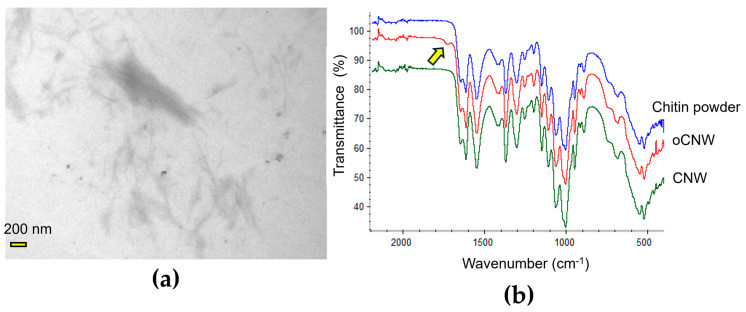
(**a**) TEM image of oCNWs, scale of 200 nm; (**b**) FT-IR spectra of pure chitin powder (blue), oCNWs (red), and CNWs (hydrolyzed) (green). Yellow arrow: peak at 1734 cm^−1^.

**Figure 3 polymers-17-00572-f003:**
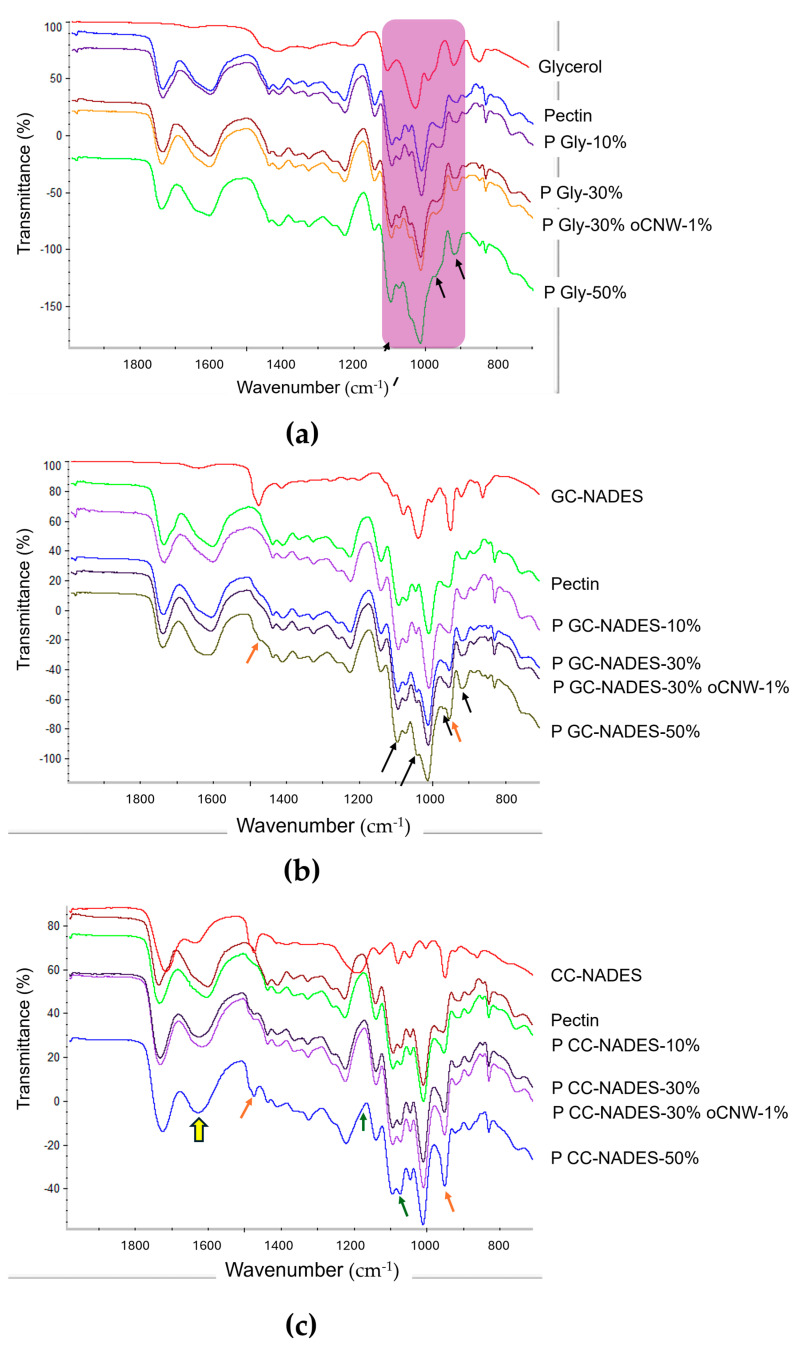
FT-IR spectra of plasticized pectin films using (**a**) glycerol, (**b**) GC-NADES, and (**c**) CC-NADES. Top to bottom: pure plasticizer; pure pectin film; and films with 10%, 30%, 30% with 1% oCNWs, and 50% plasticizer. Arrows: signals that are increased with glycerol (black), choline chloride (orange), and citric acid (green). The yellow arrow represents a signal decrease with no evidence of difference when using 30% or 50% CC-NADES.

**Figure 4 polymers-17-00572-f004:**
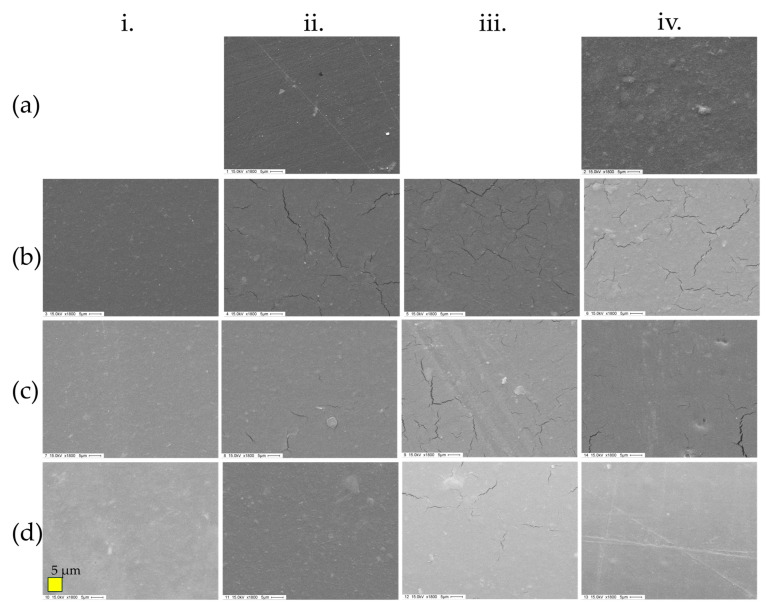
SEM images of films with (**i**) 10%, (**ii**) 30% (**iii**) 50%, or (**iv**) 30% with 1% oCNWs of (**a**) water, (**b**) glycerol, (**c**) GC-NADES, or (**d**) CC-NADES. Scale bar: 5 µm (×1800).

**Figure 5 polymers-17-00572-f005:**
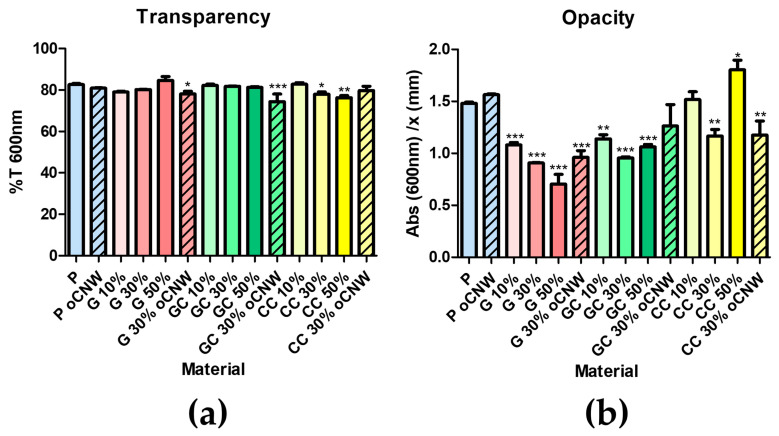
(**a**) Transparency (%T) and (**b**) opacity (Abs/*X*) of non-plasticized pectin films (blue), and those with glycerol (red), GC-NADES (green), and CC-NADES plasticizers. Striped: containing 1% oCNWs. where *, ** and *** indicate statistical significant differences with *p* < 0.05, *p* < 0.01 and *p* < 0.001 respectively.

**Figure 6 polymers-17-00572-f006:**
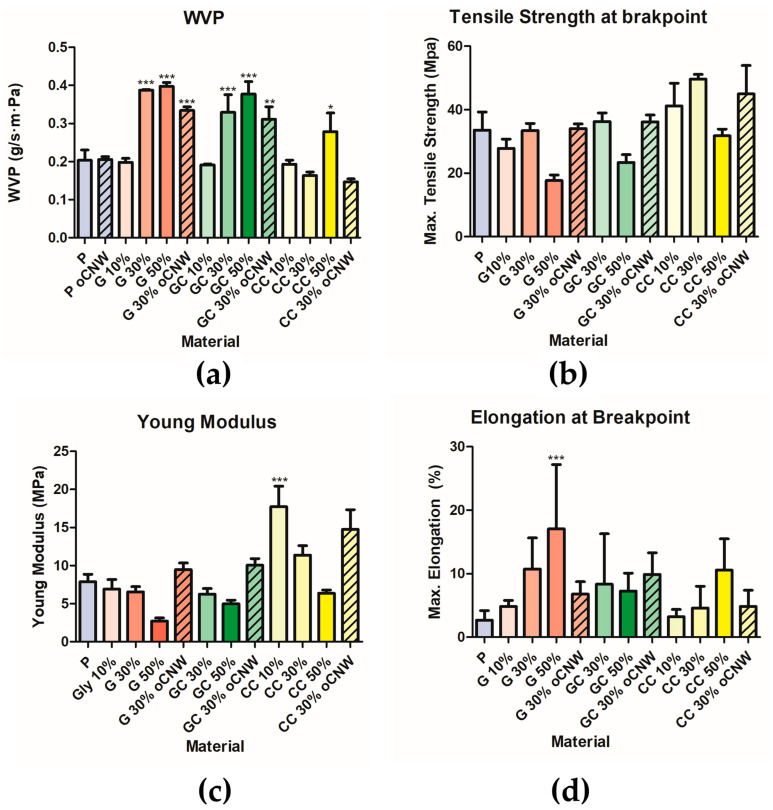
(**a**) Water Vapor Permeability, (**b**) tensile strength at break point, (**c**) Young’s modulus, and (**d**) elongation at break point from non-plasticized pectin films (blue), and films with glycerol (red), GC-NADES (green), and CC-NADES plasticizers (yellow). Striped: containing 1% oCNWs. where *, ** and *** indicate statistical significant differences with *p* < 0.05, *p* < 0.01 and *p* < 0.001 respectively.

## Data Availability

The original contributions presented in this study are included in the article and Appendix A. Further inquiries can be directed to the corresponding author.

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
