# Peer review of "Evaluation of NADES for Pectin Films Reinforced with Oxalic Acid-Modified Chitin Nanowhiskers"

_polymers, 2025, doi:10.3390/polym17050572_

Round 1

Reviewer 1 Report

Comments and Suggestions for Authors

- minor error (spelling)

- is it possible to summarize the overall characteristics of the developed modified pectin film, compared with no-modified pectin, and compare with other studies, so can see how much the improvement of mechanical strength (elongation break, young modulus)/WVP

- it seems there are variation of thickness of all samples, do you think it may affect all results (opacity and tensile strength, WVP, Young's modulus and maximum elongation)?.

- Methodology is well explained. Probably can add a flow diagram to summarize the overall research works

- It is recommended if the author can compare how much the percentage improvement compared using glycerol as plasticizer and other types of nanofiller in conclusion

- For title: recommended to write a full term for oCNW: chitin oxalate-modified nanowhiskers (as in the first glance, for me I could not capture what is it?)

Author Response

We uploaded our reply as a word file.

Reviewer 2 Report

Comments and Suggestions for Authors

Glycerol is required as a plasticiser in the manufacture of conventional edible films. This paper proposes the use of glycerol containing NADES as a plasticiser and examines different types such as hydrophobic and hydrophilic. The conclusion shows that certain NADES can be used as plasticisers in pectin films without the need of glycerol. The paper also investigates the effect of nanofillers prepared in NADESs. I think this paper is innovative and contributes to the field of edible films and DES green solvent.

Only a few suggestions:

It is recommended to use the whole name not abbrevation in the title, including keywords.

It is recommended to simplify the introduction. For example, the two paragraphs "Regarding film additives..." and "Nanofillers are another approach to improve functionality..." can be combined.

References 11-13 describe nanocomposite films as review papers. It is recommended but not necessary to add some experimental papers such as antimicrobial detailed as [antifungal gelatin-based nanocomposite films functionalized with natamycin nanoparticles] and [Antimicrobial Nanocomposite Films Reinforced with silver nanoparticles].

It is recommended to use the formula editor for the experimental formulas. Equation 3 is obviously bold and inconsistent with the two equations above.

Figure 1a is not clear. Is there a higher resolution image?

Why is oCNW prepared not just adding CNW directly in section 3.1? You can add a discussion here or in the discussion section to show the advantage of simultaneous prepare oCNW in DES and as plasticisers.

In section 3.3.3, recent studies have shown that films containing DES have a buffering effect on moisture (Buffering moisture variation during cherry tomato preservation through deep eutectic solvent films). Did the author observe this in the experiment? Or can you add remarks and discussions on related aspects?

Simplifying conclusions and list some important data.

Author Response

We have uploaded our reply to reviewer 2 as a word file.
